# A Novel Method of Transmission Enhancement and Misalignment Mitigation between Implant and External Antennas for Efficient Biopotential Sensing

**DOI:** 10.3390/s21206730

**Published:** 2021-10-11

**Authors:** Md Shifatul Islam, Asimina Kiourti, Md Asiful Islam

**Affiliations:** 1Department of Electrical and Electronic Engineering, Bangladesh University of Engineering and Technology, Dhaka 1205, Bangladesh; shifatbuet@gmail.com; 2Electroscience Laboratory, Department of Electrical and Computer Engineering, The Ohio State University, Columbus, OH 43212, USA; kiourti.1@osu.edu

**Keywords:** biopotential sensing, Fabry-Perot resonator, antenna, superstrate, metamaterials, passive sensing

## Abstract

The idea of passive biosensing through inductive coupling between antennas has been of recent interest. Passive sensing systems have the advantages of flexibility, wearability, and unobtrusiveness. However, it is difficult to build such systems having good transmission performance. Moreover, their near-field coupling makes them sensitive to misalignment and movements. In this work, to enhance transmission between two antennas, we investigate the effect of superstrates and metamaterials and propose the idea of dielectric fill in between the antenna and the superstrate. Preliminary studies show that the proposed method can increase transmission between a pair of antennas significantly. Specifically, transmission increase of ≈5 dB in free space and ≈8 dB in lossy media have been observed. Next, an analysis on a representative passive neurosensing system with realistic biological tissues shows very low transmission loss, as well as considerably better performance than the state-of-the-art systems. Apart from transmission enhancement, the proposed technique can significantly mitigate performance degradation due to misalignment of the external antenna, which is confirmed through suitable sensitivity analysis. Overall, the proposed idea can have fascinating prospects in the field of biopotential sensing for different biomedical applications.

## 1. Introduction

Biopotentials are electric signals which are generated inside a biological substance through the electro-chemical activities of a cluster of cells. Through the sensing of biopotentials, it is often possible to describe the physiological state of a group of cells which are of interest. These biopotentials are extremely weak, with magnitudes in the order of μV [1]. The biosignals are often modulated by a higher frequency for efficient transmission, and this modulation frequency ranges from MHz to GHz [2]. In general, almost all biosensing tools involve measuring electrodes, whose task is to exploit the electrochemical activities of the cell and transform to equivalent voltage signals for detection.

Depending on the extraction mechanism of the signal, sensing can be either “active” or “passive”. In active sensing, biosignals are processed and received directly by external devices, which can be different integrated circuits (ICs), harvesters, and sensors. In the more recently explored passive sensing, there is an implanted/interrogator antenna pair that communicates signals between the two. The implanted antenna is connected to the electrode and is buried inside the tissue, while the interrogator antenna is an external antenna to which the biosignals are transmitted. The passive sensing mechanism has been tried on different biopotential sensing systems [3,4,5,6]. The main advantage of the passive sensing systems compared with the active sensing systems is that passive systems do not require obtrusive circuitry, and such wireless links are easily portable and wearable. In passive sensing, typically, the two antennas are placed close to each other [3,4,5,6,7] so that they can inductively couple with each other. This requires the two antennas to be separated by a small distance, leaving almost no room for engineering improvements that could boost transmission. By contrast, since the antennas are tightly coupled, change in the dimension or parameters of one antenna directly influences the behavior of the second antenna. This makes the passive sensing systems both difficult to engineer for transmission enhancement and susceptible to misalignment, which is a very likely event if the system is used as a wearable device [3]. Most of the antennas in the literature for passive sensing rely on near-field coupling, and the associated issue of misalignment remains largely unaddressed to date.

On the other hand, not related to such passive sensing application, a significant amount of work has been done to enhance antenna gain for efficient transmission in the far field region. Theoretical and experimental methods include the use of reflecting planes [8,9], use of superstrate dielectric [10,11], and metamaterials with or without superstrates [12,13,14,15,16,17,18]. At first, ‘metamaterial’ was defined as periodic structures capable of showing negative electromagnetic properties [19,20,21,22], but, today, any external tampering with materials which results in the change of effective electromagnetic properties is termed as metamaterial [23,24]. The enhancement of directivity in the aforementioned works is mainly based on one of the following three principles: (a) the model of Fabry-Perot resonator cavities, where the superstrate or the reflecting planes induce multiple reflections inside the antenna ground-superstrate gap [8,9,10,11,15,16,25], (b) the idea of negative refraction and inverse focusing, which is induced by left-handed materials (LHM) [12,13,14], and (c) the ability of different periodic structures to induce a small frequency band just above the resonant frequency, where the refractive index is less than that of air [18,20,21,22]. Published results in these works have shown considerable enhancement in the directive gain of the antenna.

While none of these works directly provide any insight on transmission between antenna pairs for passive sensing, it is, of course, encouraging to try similar concepts and see how they contribute to the problem of interest. However, attempting these concepts for passive biosensing is not straightforward. For example, both the principles (b) and (c) mentioned above are highly dependent on operating frequency and work in a very small frequency band. Since the passive sensing antennas are closely coupled with each other and the resonant frequency changes with small misalignment and fabrication imperfections, it is indeed challenging to design structures that operate at a particular frequency (or in a small band). Furthermore, the direction of periodicity and the period along the propagation direction requires large spatial requirement, and complex fabrication, which is not desirable in passive sensing. These leave idea (a) (Fabry-Perot resonators) to have some probable positive effect on transmission enhancement. The merit of this idea is that the governing principle only depends on the reflective nature of the designs, not on frequencies, and is, therefore, more convenient to implement. While we focus on this approach to enhance transmission, there is also a major hurdle to overcome. To incorporate the superstrate and the metamaterials in between the antenna pairs, the separation between them needs to be increased. However, increasing this separation reduces the transmission between the antenna pair. So, the gain in performance has to surpass the loss of transmission due to increasing the antenna pair separation, ultimately achieve an overall increase in transmission, and, of course, fill up the transmission requirement of the desired application.

In this work, we propose a new technique of increasing transmission between the implanted/interrogator antenna pair for biopotential sensing, adopting the concept of Fabry-Perot resonators and metamaterials. The obtained transmission results are compared with the existing literature, to demonstrate the method’s efficacy. In addition, a sensitivity analysis is carried out, and it is shown that the transmission performance remains nearly unaltered for reasonable misalignment between the two antennas.

The rest of the paper is divided into four sections. In Section 2, we present the concept of the Fabry-Perot resonator and the motivation of our proposed approach. In Section 3, we describe two simulation environments where we test our ideas and observe the transmission characteristics. In Section 4, we perform a simulation and subsequent analysis with practical materials and biological media models to demonstrate a passive biosensing application. We summarize the work with final remarks in Section 5.

## 2. Prior Art

The method of improving the directive gain of an antenna with an ideal (infinitely large) ground was first mathematically explored in Reference [8]. There, a partially reflecting sheet was placed above the infinite antenna ground plane at a distance $\frac{\lambda }{2}$ to form a resonant cavity. The sheet has a planewave reflection coefficient of magnitude *R*. The cavity enhances the gain of the antenna by a maximum factor of:(1)$$gain=\frac{1+R}{1-R}.$$

This formula suggests that the more reflective the sheet is, the higher the gain enhancement becomes. A study, where multiple superstrate materials were used instead of the sheet [10], also agrees with the idea. However, in this approach, there are two issues:A physical approximation of the infinite ground requires a large antenna plane, which is not feasible for small domain applications, such as biosensing. Therefore, we need to investigate what the superstrate can offer in transmission for finite size grounds.The replacement of the reflecting sheet with the superstrate adds another unknown, which is the thickness of the superstrate, which is needed to be optimized for practical and finite sized patches.

In a subsequent work [15], it has been shown that the inclusion of metamaterials on the superstrate surface can enhance the reflectivity; therefore, the superstrate will demonstrate higher reflection than the material itself can offer. Later, in Reference [25], the effect of metallic imprints on gain enhancement was examined in further detail for circular patch antennas, and it has been shown that the gain enhancement is not indefinitely proportional with the surface reflection as Equation (Equation 1) would suggest, and the idea of “optimum reflection coefficient” was introduced. The claim was that, up to a certain value of superstrate reflection, gain will increase, and then the gain will decrease very rapidly. So, the task of the metamaterial designer is to make the meta imprinted superstrate acquire that optimum reflection, for which maximum gain will be achieved.

Since, in this work, we investigate the same concept on the transmission characteristics between two antennas, the above discussions boil down to the following expectations:With the increase of superstrate permittivity, transmission will enhance but only up to a given maximum superstrate permittivity.Insertion of metamaterial imprints on low permittivity superstrates can produce the optimum performance at a higher permittivity. This tool can be specially useful since practical high permittivity materials are not always commercially available, and they provide relatively large dispersive loss.

Along with exploring these ideas, we also propose and investigate the effect of filling the patch-superstrate gap with dielectric materials. The idea came up with two motivations:The insertion of the dielectric material turns the gap into a dielectric waveguide channel, where some of the leaky waves can reflect into the channel and increase the number of reflections (see Figure 1b), and possibly overall transmission.The effective wavelength distance will be $\frac{\lambda }{2\sqrt{\epsilon_{gap}}}$, $\epsilon_{gap}$ being the permittivity of the gap filled material. Hence, the system can be miniaturized, which is always desirable in implementation of biosensing systems.

## 3. Examining Transmission Enhancement

In this section, we present simulation analysis on the performance enhancement in transmission due to variation in each variable, namely the dielectric properties of the superstrate and the gap insertion material, and the effect of metamaterials. First, we describe the simulation environment in which we will explore the effects, and then provide the analysis. Throughout the work, the frequency of interest is 4.8 GHz, similar to the implant operating frequency in Reference [3].

### 3.1. Simulation Systems

For our analysis, we will explore two different antenna pair systems and examine transmission characteristics on each of them. The first system is the simpler one, where the two antennas are placed in free space, whereas, for the second system, one antenna is placed in air, and the second antenna is placed in a medium other than air. This new medium has an assumed dielectric constant of $\epsilon_{tissue}=40$ and a conductivity of $\sigma_{tissue}=3$ Sm${}^{-1}$ at 4.8 GHz. We choose these values for simulation because most biological tissues have dielectric properties of similar range [26]. For clarity’s sake, we denote these two systems as “system 1” and “system 2”, respectively (see Figure 2).

For each of the two systems, we analyze the transmission characteristics in the following steps:*Step 1:* First, we take note of the transmission loss for the two systems, without any engineering in between. For system 1, the two antennas are separated by a wavelength distance in air $\lambda_{air}\approx 62.5$ mm at 4.8 GHz For system 2, the medium boundary is placed at a distance $\frac{3\lambda_{air}}{4}\approx 46.875$ mm from the external antenna. The internal antenna is buried further $\frac{\lambda_{tissue}}{2}=\frac{\lambda_{air}}{2\sqrt{\epsilon_{tissue}}}\approx 4.94$ mm inside (see Figure 2a).*Step 2:* Since the air-superstrate cavity needs to have a half-wavelength distance, we place a superstrate material at a distance $\frac{\lambda_{air}}{2}\approx 31.25$ mm above antenna 1, for both systems. The location of the antennas will remain unchanged (see Figure 2b). From there on, we vary the superstrate dielectric constant ($\epsilon_{sup}$) and take note of the transmission characteristics.*Step 3:* Next, we fill the gap between the patch and the superstrate with a dielectric having $\epsilon_{gap}=2$. To maintain the same half wavelength cavity, the new air-superstrate gap is now $\frac{\lambda_{gap}}{2}=\frac{\lambda_{air}}{2\sqrt{\epsilon_{gap}}}\approx 22.09$ mm (see Figure 2c). In this way, although the distance between the antenna and the superstrate is reduced, the overall distance is the same if measured in wavelengths. We again vary the dielectric constant of the superstrate to observe the transmission performance.*Step 4:* Finally, we insert circular ring metamaterial imprints on both sides of the superstrate for a relatively low $\epsilon_{sup}=3$ to induce high permittivity substrate transmission from the previous step.

### 3.2. Antenna Pairs

Throughout the analysis, we have noted that the antenna frequency change in each of the four steps mentioned above, and, for proper comparison at the desired frequency, we tune the antenna pair regularly. Therefore, we have used simplified patch antennas, which could be tuned to the desired frequency of operation rather easily using parametric analysis.

Both of the two antenna substrates are assumed to be made of Rogers RO4003 material ($\epsilon_{r}=3.55,tan\delta =0.0027$), with thickness of 0.762 mm. The width of the feedline is 0.5 mm, and the length of the feedline is 5.24 mm (see Figure 3a). For both the antennas, the substrate has the dimension 2W×2L. The parameters of the two antennas, in each of the steps to tune at the desired frequency, are given in Table 1. L1,W1 and L2,W2 are the L,W values for antenna 1 and 2, respectively.

In addition, for system 2, we have also covered the implant patch (antenna 2) with a 1-mm thick coverage of lossless dielectric material ($\epsilon_{r}=2$). It has been shown that such coating not only ensures biocompatibility but also improves the transmission significantly by reducing dispersion loss inside the lossy medium [3,27]. To illustrate this, the amount of transmission with and without the presence of the coating is shown in Figure 4, from which transmission enhancement of ≈14 dB is observed when insulation is added.

The metamaterial imprints are consisting of simple two concentric rings, the reflectivity is modified through parametric analysis by increasing/decreasing the ring width *c* (see Figure 3b).

### 3.3. Transmission Analysis

In this section, we present the simulation results for each of the two systems mentioned before. For the analyses, we use two widely used EM simulation tools: ANSYS HFSS and CST microwave studio. In the following discussions, we present the simulation results from HFSS, whilst mentioning the CST verified results in parenthesis.

For system 1, the initial transmission loss is found to be −9.33(−9.30) dB. After adding the superstrate, at first the performance improves with the increase of superstrate permittivity. The best transmission of −5.54(−5.67) dB is found for $\epsilon_{sup}=11$, which is an improvement of ≈3.8 dB (see Figure 5a). After $\epsilon_{sup}>11$, we see a fast reduction in transmission. Therefore, we identify the maximum limit of superstrate reflection (permittivity) for which the transmission is maximum. Next, after filling up the patch-superstrate gap with the dielectric, we see that the transmission improves further, for all values of superstrate dielectric constant. The best transmission is found to be −4.08(−3.78) dB for $\epsilon_{sup}=5$. Later, after adding circular rings on both sides of the superstrate, we obtain a similar transmission performance of −3.96(−3.63) dB for $\epsilon_{sup}=3$ (see Figure 5b). From the final transmission curves, we find that an improvement of more than 5 dB has been achieved after all these steps.

For system 2, the initial transmission loss after the inclusion of insulation layer was found to be −24.91(−27.2) dB (see Figure 3). From there on, the performance improved with the inclusion of superstrates, and reached a maximum of −21.7(−23.9) dB for $\epsilon_{sup}=5$ (see Figure 6a). Further increase of $\epsilon_{sup}$ degraded transmission. Later, after filling the patch superstrate gap with dielectric material, the transmission improved yet again. The maximum transmission was noted to be −17.06(−19.36) dB for $\epsilon_{sup}=5$. Finally, after including circular rings on the superstrate, we obtain a transmission of −17.11(−18.53) dB for $\epsilon_{sup}=3$ (see Figure 6b). Therefore, a total transmission enhancement of ≈8 dB has been achieved for system 2.

Simulations using both HFSS and CST agree with almost identical amount of gain enhancement through all the steps. However, the simulated transmission losses differ slightly for the two softwares: by ≈0.3 dB in system 1, and ≈2 dB in system 2. This can be attributed to the difference between the difference in the solver systems and excitations.

After exploring the effects of superstrate, gap filler, and metamaterials for different systems, we can draw the following conclusions:With the inclusion of the superstrate material, the transmission improves, but only up to a certain limit.Filling up the patch-superstrate gap with dielectric material further enhances the transmission.

## 4. Implementation: A Passive Neurosensing System

In this section, we design a passive neuropotential recording system [3,4,7] as a potential application of biosensing with the developed method. In the design, the biological media consist of a layered structure of skin, bone, brain grey matter, and brain white matter [4]. Each of these tissues are frequency dispersive, whose dielectric properties are presented in Figure 7. In the arrangement, the implant antenna is placed 1 mm inside the skin tissue. The external antenna is placed 3 mm outside the skin to form the passive sensing system.

### 4.1. Antenna Design

As in the previous section, the two antennas here are simple patch antennas, and tuned to operate at 4.8 GHz (see Figure 3a). As antenna substrate, Rogers RO4003 dielectric has been used, with thickness of 0.762 mm Each of the conducting surfaces (patch, ground, circular rings) of the arrangement is coated with PDMS ($\epsilon_{r}=2.8,tan\delta =0.002$) with a chosen thickness of 1.5 mm.

For the two antennas, L1=L2=15.24 mm, W1=W2=19.31 mm. The length of the feed (feed_len) is 5.24 mm, and the width (feed_wid) is 0.5 mm. The dimension of the external and implant antenna substrates are 2W1×2L1 and 1.2W2×1.2L2, respectively.

Rogers RO4003 has been used as the superstrate material with a thickness of 8.5 mm. Metamaterials are printed on both sides of the superstrate. The gap between the external patch and the superstrate is filled with PDMS material which is 18.675-mm thick. The metamaterial imprint is similar to the circular rings in Figure 3b, where we choose r=2.1 mm, c=0.8 mm, and d=0.2 mm. A sectional view of the whole arrangement is presented in Figure 8a.

### 4.2. Performance Analysis

After designing a biopotential recording system with the proposed concept, we simulate the transmission characteristics between the two antennas. The simulated transmission plots are shown in Figure 8b. At the resonant frequency of 4.8 GHz, the system shows a transmission loss of −11.27 dB in the simulation in HFSS (−10.50 dB in the simulation in CST). This transmission performance compares very well with some of the existing works as presented in Table 2. The dimension of the is also comparable with what has been reported in those works. However, the inclusion of superstrate and filling material makes the proposed design slightly more bulky and complex compared to those.

An important analysis for such neurosensing applications is the performance degradation of the system under possible misalignment of the external antenna. For the proposed antenna system, we also explore this performance degradation misalignment from the center position. The proposed system appears to be very stable with the misalignment along both the *x* (along the feedline) and *y* axis (perpendicular to the feedline) (see Table 3), with maximum performance degradation of only 2.41 dB, for 1 cm offset.

Another key consideration is the implementation of the proposed concept requires a layered geometry with fine tuned parameters, and a slight change in any of the thicknesses (superstrate, dielectric fill, air gaps, etc.) should also influence the performance of the design. Therefore, we present a short sensitivity analysis with the change of these parameters. For the dielectric gap between the source and the superstrate, a 1 mm increase (decrease) of the dielectric thickness degrades the performance by 0.1(0.48) dB. For the superstrate, similar increase (decrease) of the thickness decreases transmission by 0.15(0.15) dB. Such low performance degradation is encouraging as in practice, it may not always be possible to assemble the system with strict dimension requirements. However, maintaining a 3 mm air gap may also be challenging, and, for the gap thickness, a 1 cm increase results in a more prominent degradation of around 5.9 dB. This is comparatively large, but the systems still maintains a very high transmission performance.

## 5. Conclusions

A new method of transmission enhancement and misalignment mitigation between internal and external antenna pair for biopotential sensing is proposed in this paper. A full-wave simulation of a typical neuropotential system is carried out, taking into account the realistic permittivity and conductivity of biological tissues, along with proper modeling of the antenna pair. Exploiting the theory of Fabry-Perot resonators and metamaterials, the proposed technique achieves a very high transmission gain of around −11 dB. The design also appear to be very stable under possible misalignment of the antenna pairs, and small dimension mismatch from the optimum design. The proposed technique is anticipated to find its application in the area of biosensing for the diagnosis, monitoring, and treatment of several critical health conditions. 

## Figures and Tables

**Figure 1 sensors-21-06730-f001:**
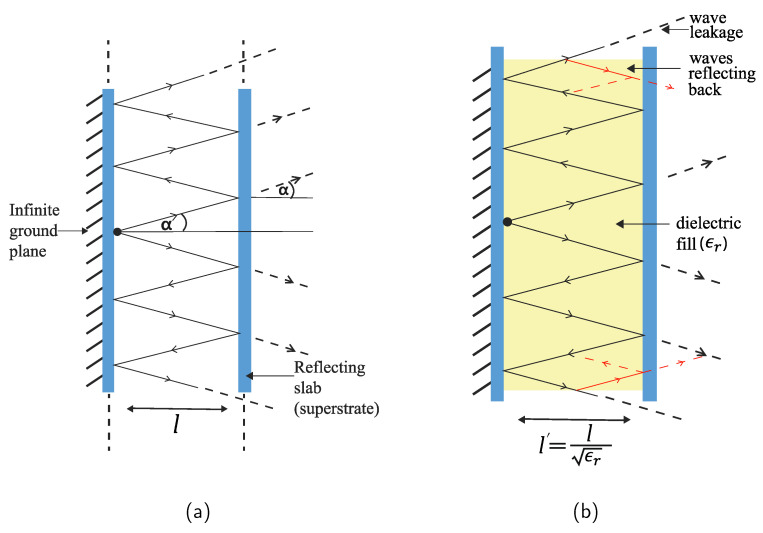
Geometric interpretation of the Fabry-Perot resonator antenna concept. (**a**) Waves leak from the top and the bottom of the cavity. (**b**) Leaked waves are reflected back to induce further reflection.

**Figure 2 sensors-21-06730-f002:**
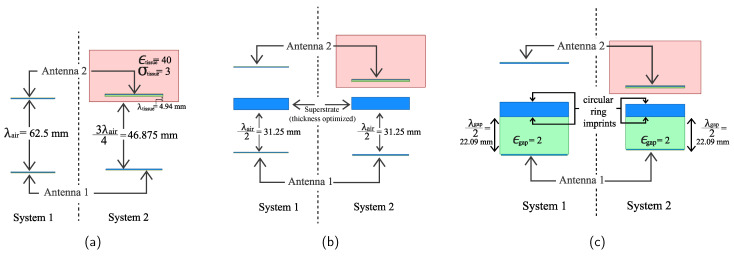
Step-by-step schematic diagram for the analysis of the two defined different antenna systems. (**a**) The antenna pair is placed without any engineering in between. (**b**) Superstrate material is inserted at a distance $\frac{\lambda_{air}}{2}$ in front of antenna 1. (**c**) The gap between antenna 1 and the superstrate is filled with a dielectric of $\epsilon_{gap}=2$.

**Figure 3 sensors-21-06730-f003:**
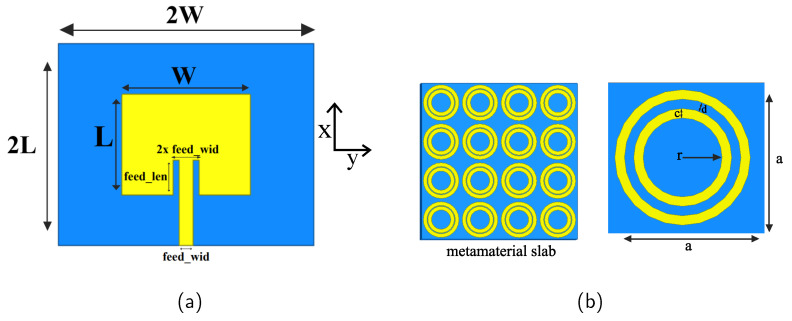
(**a**) The common geometry of the two antenna configurations for the analysis. (**b**) The view of circular ring imprints on both sides of the superstrate with tunable parameters.

**Figure 4 sensors-21-06730-f004:**
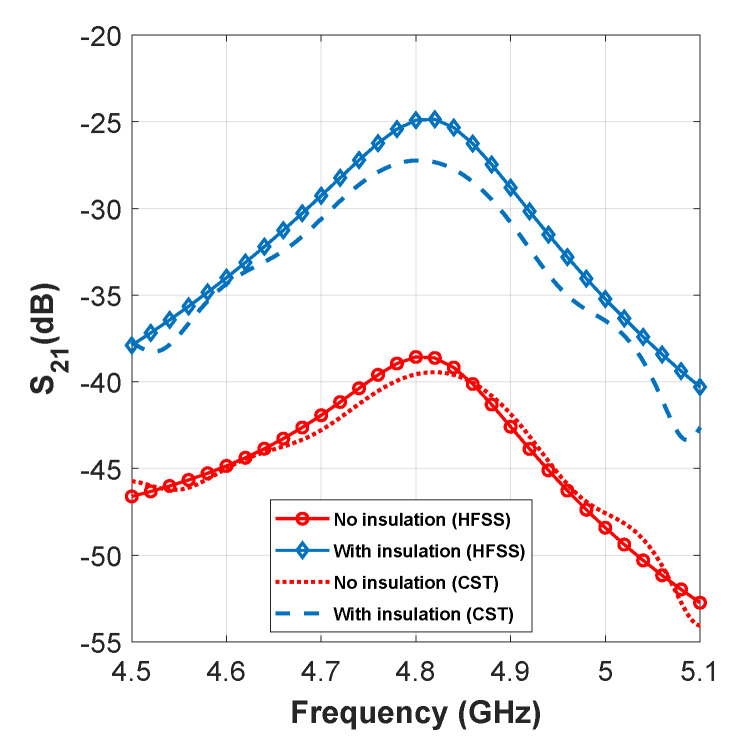
Effect of insulation to enhance transmission gain.

**Figure 5 sensors-21-06730-f005:**
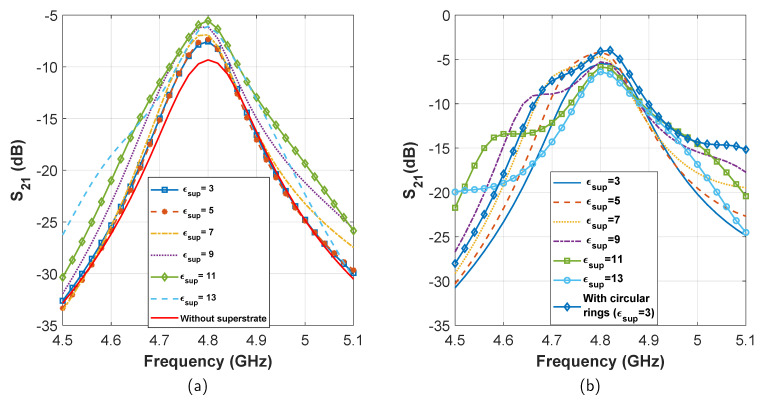
Effect of transmission enhancement with varying superstrate dielectric constant in system 1. (**a**) Gain enhancement in steps 1 and 2. (**b**) Gain enhancement in steps 3 and 4.

**Figure 6 sensors-21-06730-f006:**
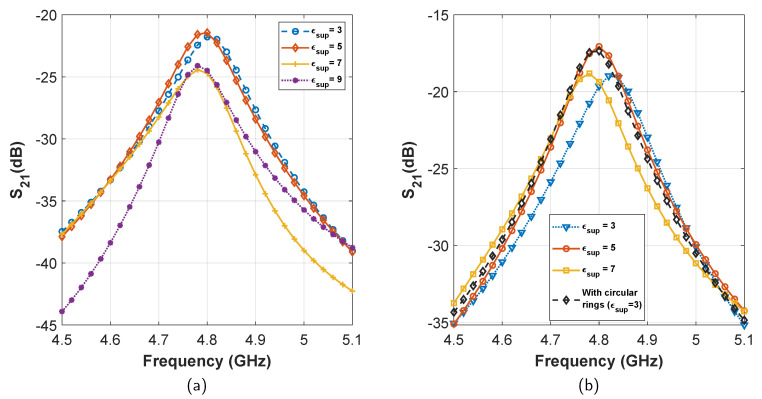
Effect of transmission enhancement with varying superstrate dielectric constant in system 2. (**a**) Gain enhancement in steps 1 and 2. (**b**) Gain enhancement in steps 3 and 4.

**Figure 7 sensors-21-06730-f007:**
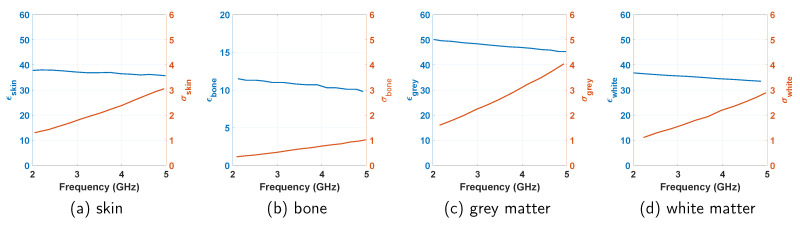
Frequency dependent dielectric constant (ϵ) and conductivity (σ) of the tissues to design the neurosensing system.

**Figure 8 sensors-21-06730-f008:**
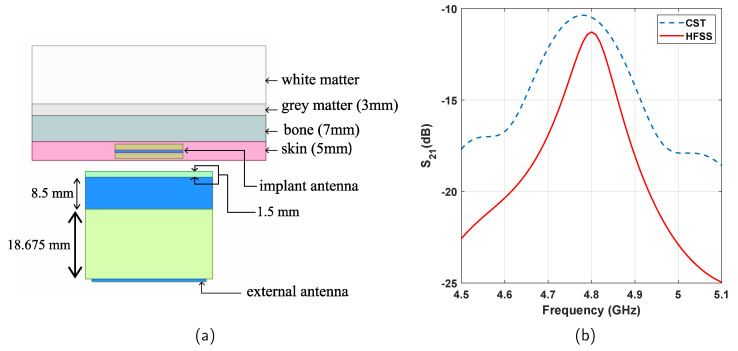
(**a**) Sectional view of the proposed neuropotential recording system. (**b**) Transmission performance of the design.

**Table 1 sensors-21-06730-t001:** Dimensions of the two antennas for each system, and for each step of the analysis. All units are in mm.

	System 1	System 2
**Step 1**	L1=15.97, W1=20.21	L1=16.03, W1=20.29,
	L2=15.97, W2=20.21	L2=15.87, W2=20.29
**Step 2**	L1=15.97, W1=20.21	L1=16.03, W1=20.29,
	L2=16.03, W2=20.29	L2=15.87, W2=20.29
**Step 3**	L1=15.4, W1=19.5	L1=15.71, W1=19.88,
	L2=16.03, W2=20.29	L2=15.87, W2=20.29
**Step 4**	L1=15.45, W1=19.5	L1=15.71, W1=19.88,
	L2=16.03, W2=20.29	L2=15.87, W2=20.29

**Table 2 sensors-21-06730-t002:** Comparison of the performance of the proof of concept design with some of the existing biosensing works.

Reference	Operating Frequency	Maximum $S_{21}$	External Antenna Surface	Implant Antenna Surface
[3]	≈4.8 GHz	−19 dB	145 mm diameter	$15\times 16\;{\mathrm{mm}}^{2}$
		(−17 dB in simulation)		
[28]	2.4 GHz	−22.5 dB	$12\times 12\;{\mathrm{mm}}^{2}$	same as the external
			$9\times 10\;{\mathrm{mm}}^{2}$, etc.	antenna
[29]	2.4 GHz	−21.4 dB	$24.9\times 24.9\;{\mathrm{mm}}^{2}$	N/A
[30]	≈4.8 GHz	−18.2 dB	$40\times 40\;{\mathrm{mm}}^{2}$	$19.94\times 29.17\;{\mathrm{mm}}^{2}$
[31]	400 MHz (2.4) GHz	−33 dB	$22\times 23\;{\mathrm{mm}}^{2}$	N/A
[32]	400 MHz	−24 dB	$26.8\times 28\;{\mathrm{mm}}^{2}$	N/A
proposed design	4.8 GHz	−11.27 dB (in HFSS)	$30.48\times 38.62\;{\mathrm{mm}}^{2}$	$18.29\times 23.172\;{\mathrm{mm}}^{2}$
		−10.50 dB (in CST)		

**Table 3 sensors-21-06730-t003:** Performance degradation of the proof of concept design with misalignment of the external antenna.

Misalignment along *x*		Misalignment along *y*	
Axis		Axis	
Distance (mm)	$S_{21}$ degradation (dB)	Distance (mm)	$S_{21}$ degradation (dB)
−10	2.41	−10	1.11
−5	0.67	−5	0.14
5	0.62	5	0.14
10	2.33	10	1.11

## Data Availability

Not applicable.

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
