# Peer review of "A Novel Method of Transmission Enhancement and Misalignment Mitigation between Implant and External Antennas for Efficient Biopotential Sensing"

_sensors, 2021, doi:10.3390/s21206730_

Round 1

Reviewer 1 Report

Submitted manuscript is interesting, adding knowledge to a long investigated problem. Measurements are missing, so verification cannot be actually realised. The difference between CST and HFSS of 2dB, it is high but would not be a problem if authors would not be based on more or less 3db in order to support their claim of better design. Measurements usually solve such issues. 

I would comment that in section 2 a missing word is altering the message conveyed. I would say that authors are not using a reflecting sheet but a partially reflecting. A reflecting sheet would have different results

Also it seems that authors consume space for the "theory" when actually the message is λ/2. They justify the time spent because of missing justification in other publication. This is half true, still the writting is useful, I would decrease the length of writting.

Secondly I do not understand why simulations are being carried out in order to "explore" the concept of enhancement and then a design is proposed which is actually the same as the one used before. I do not see why this is needed

Finally, authors mention that they are in far field. Can authors explain why they are in the far field? This is based on which numbers and theory? I understand that there is not much degradation with misalignment but this does not mean far field radiation necessarily 

Author Response

Thank you for the careful review of the paper. We tried to address all the issues and believe that the quality of the paper has been improved significantly. The response letter is attached for your consideration.

Reviewer 2 Report

In this manuscript, the authors developed a novel method of transmission enhancement and misalignment mitigation between implant and external antennas for efficient biopotential sensing. The work is well written and structured, sufficiently innovative in the field of biopotential sensing. I suggest minor revision before publication as follows:

1 The icon’s annotations in fig 5(b) and fig 6(b) block part of the curves, please adjust the position to make the data more complete and clearer.

2 Please balance the size of the pictures in figure 8 and adjust their position.

3 Line 254 and line 263,the blank is too long.

4 You should balance the font size in all figures, as large as the main body. For example, the characters are too small in Fig 1 and 2, and the font size is inconsistent in Fig 3 and 8. The quality of figures should be further improved. For example, Fig 8 is quite messy without keeping in alignment.

5 Line 50, “Although the term ‘metamaterial’ was introduced to define artificial” you should replace “metamaterial” by “artificial”.

Author Response

(The authors gave the same response as above.)

Round 2

Reviewer 1 Report

Authors have responded very well. Measurements is still an issue, I do not agree they not needed  but Ι understand the scope of manuscript and the context of journal

Two comments though:

1) Commenting on the boundary conditions used in HFSS is unneeded as this is trivial. This does not prove much and as said the deviation between the two different numerical techniques would be solved by measurements

2) Authors provide a sensitivity study about misalignment. I would say that in the present case, deviation in z axis would be useful as well. It is curious that simulated thickness of materials needs to be that exact and I am wondering what would happen if this changed (as it is expected to change) 
